# Impact of Dietary Arachidonic Acid on Gut Microbiota Composition and Gut–Brain Axis in Male BALB/C Mice

**DOI:** 10.3390/nu14245338

**Published:** 2022-12-15

**Authors:** Katleen Pinchaud, Zeeshan Hafeez, Sandrine Auger, Jean-Marc Chatel, Sead Chadi, Philippe Langella, Justine Paoli, Annie Dary-Mourot, Katy Maguin-Gaté, Jean Luc Olivier

**Affiliations:** 1Calbinotox (UR7488), Université de Lorraine, 54000 Nancy, France; 2INRAE, Université Paris-Saclay, AgroParisTech, UMR 1319 Micalis Institute, 78352 Jouy-en-Josas, France; 3CHRU de Nancy, Pôle des Laboratoires, Service de Biochimie-Biologie Moléculaire-Nutrition, 54000 Nancy, France

**Keywords:** arachidonic acid, hyperlipidic diet, gut microbiota, gut–brain axis, inflammation

## Abstract

Although arachidonic acid (ARA) is the precursor of the majority of eicosanoids, its influence as a food component on health is not well known. Therefore, we investigated its impact on the gut microbiota and gut–brain axis. Groups of male BALB/c mice were fed either a standard diet containing 5% lipids (Std-ARA) or 15%-lipid diets without ARA (HL-ARA) or with 1% ARA (HL + ARA) for 9 weeks. Fatty acid profiles of all three diets were the same. The HL-ARA diet favored the growth of *Bifidobacterium pseudolongum* contrary to the HL + ARA diet that favored the pro-inflammatory *Escherichia–Shigella* genus in fecal microbiota. Dietary ARA intake induced 4- and 15-fold colic overexpression of the pro-inflammatory markers IL-1β and CD40, respectively, without affecting those of TNFα and adiponectin. In the brain, dietary ARA intake led to moderate overexpression of GFAP in the hippocampus and cortex. Both the hyperlipidic diets reduced IL-6 and IL-12 in the brain. For the first time, it was shown that dietary ARA altered the gut microbiota, led to low-grade colic inflammation, and induced astrogliosis in the brain. Further work is necessary to determine the involved mechanisms.

## 1. Introduction

Dietary lipids are the main source of energy and represent between 10 and 58% of daily energy intake in humans [1]. Beyond this quantitative aspect, the composition of dietary fat, particularly the distribution of various fatty acids, strongly influences health and well-being. Since the beginning of the 2000s, many research works and nutritional surveys have focused on the ω-6/ω-3 polyunsaturated fatty acids (PUFAs) ratio [2,3,4]. In the last decades, an increase in the dietary intake of ω-6 PUFAs has been observed with the westernization of diets throughout the world. The current data are, therefore, far from the recommended value of 4–5 for the ω-6/ω-3 ratio [5]. In the majority of foods, ω-6 and ω-3 PUFAs are provided in the form of their shortest precursors, linoleic (LA) or α-linolenic (ALA) acids, respectively. The longest ω-6 PUFA, arachidonic acid (ARA), and ω-3 PUFAs, eicosapentaenoic (EPA) and docosahexaenoic (DHA) acids, are synthesized through the elongation and desaturation of their precursors or directly provided by the consumption of marine or animal products. Regarding the dietary intake of ARA and DHA, international evaluations have shown that there are large variations worldwide [6]. Many countries do not meet the nutritional recommendations [6]. Dietary ARA and DHA are partially incorporated into cell phospholipids. The influence of the dietary ω-6/ω-3 ratio on human well-being can be explained by respective pro- and anti-inflammatory activities of the ARA and DHA/EPA derivatives, which are synthesized after the phospholipase A_2_ (PLA_2_)-mediated release of these PUFAs from cell phospholipids [7].

Numerous works have shown that dietary ω-3 fatty acids deficiency is involved in various pathologies from atherosclerosis and cardiovascular pathologies [8] to metabolic [9] and neurodegenerative diseases [10]. Despite this, the influence of excessive intake of ω-6 PUFAs on human well-being is more debatable, as the number of studies on this issue, particularly the role of ARA, are rather low. Nevertheless, some epidemiological studies indicate a correlation between higher incorporation of ARA in erythrocyte phospholipids and pathological features of accelerated aging such as reduction in telomeres length and cognitive impairments [11,12]. However, Zhuang et al. [13] contrarily suggested positive effects of a higher proportion of ω-6 PUFAs on mortality in American and Chinese populations and even stated a lower mortality risk with a ω-6/ω-3 ratio between 6–10 in Chinese people. Studies have shown that distribution of ω-6 PUFAs and their effects on inflammation levels and associated pathologies can be influenced by polymorphisms in several genes involved in their synthesis and release from phospholipids [14,15,16]. Besides these epidemiological studies, preclinical works suggested negative impacts from dietary ARA intake. We previously observed that a 10%-fat diet supplemented with 1% ARA worsened the neurotoxicity of intracerebro-ventricle-injected amyloid-β1–42 peptide in mice, which is the main agent of Alzheimer’s disease [17]. At first glance, one can assume that this negative effect could be due to the accumulation of ARA in the brain or its conversion into pro-inflammatory eicosanoids. However, this ARA-rich diet highly increased ARA levels in the liver and erythrocytes compared with the brain where ARA incorporation was moderate. This could barely support mechanisms based on the direct effect of intracerebral ARA. Supplementation of 1% ARA in a high-fat diet exacerbated obesity in rats and reduced bone mineral content [18]. However, this work did not mention the ARA accumulation in bone or other tissues. In contrast, Boyd et al. [19] recently demonstrated that ARA and LA accumulate in lumbar dorsal root ganglia after excessive intake of an ARA-enriched high-fat diet. An increase in phospholipase A_2_ activity consecutively led to the release of ARA, nociceptive behavior, and allodynia. In the Danio rerio model, a diet supplemented with ARA modified the expression levels of several genes involved in immune function and eicosanoids synthesis, especially after *Streptococcus* infection [20,21]. Interestingly, in this zebrafish model, an ARA supplemented diet increased the gut microbiota diversity and expression of eicosanoids-producing genes such as cyclooxygenase-2 in gut mucosa [22].

Gut dysbiosis has been described in various systemic pathologies such as atherosclerosis, obesity, and type II diabetes [23,24]. Furthermore, numerous works underlined the role of the microbiota–gut–brain axis in neurodegenerative diseases such as Alzheimer’s and Parkinson’s diseases [25,26], in autism spectrum disorders [27], as well as neuropsychiatric affections such as depression [28]. While etiology of these various pathologies may vary, the bidirectional relationship between gut and brain influences their development and prognosis and is putatively based on several mechanisms, especially in neurodegenerative diseases [29]. These mechanisms include direct neural communications through the vagal system or indirect metabolic, endocrine, or immune communication through passages of bacterial components such as lipopolysaccharides (LPS), microbial metabolites such as short chain fatty acids, secretion of pro-inflammatory cytokines, or neuroactive molecules [30]. Among these various mechanisms, alterations of the intestinal barrier and associations of systemic inflammation and neuro-inflammation play critical roles [31]. For example, inflammatory bowel disease constitutes a risk factor of Alzheimer’s disease [32] and the fatty acids composition of a diet modulates microglial cells activities in colitis model mucin-2 knock-out mice through modifications of interleukin-2 and interferon γ expressions in the brain [33]. Since diet can be responsible for 60% variations of gut microbiota [34], dietary intervention can be integrated in the therapeutic strategies against neurodegenerative diseases as well as other brain affections in which the gut–brain axis is involved. Contrary to the polysaccharides-rich diet [35], little is known about the influence of dietary fatty acids on the gut microbiota. Fatty acids are supposed to integrate into the small intestine. However, an increased intake of dietary fatty acids can indirectly influence the bile acid fluxes and, consequently, modulate the microbiota composition [1]. Furthermore, several works demonstrated that ω-6 fatty acids exert specific effects on gut mucosa and microbiota. Ghosh et al. [36] found that high-fat diets enhanced the association of microbiota with ileal mucosa of 2-year-old mice but only an ω-6-(LA)-rich diet caused bacterial and neutrophil infiltrations of the mucosa. More recently, Selmin et al. [37] observed that the administration of a ω-6-rich soybean-oil-based diet containing 20% lipids (*w*/*w*) led to a pathobiontic profile in the gut of male C57BL/6j mice compared with a diet containing 11% lipids (*w*/*w*), mostly saturated fatty acids. Mice fed a soybean-oil-based diet showed a reduction in the presence of *Firmicutes*, *Clostridia*, and *Lachnospiraceae* which was counterbalanced by an increase in the abundance of *Bacteroidetes* and *Deferribacteraceae*. These microbiota modifications were associated with cyclo-oxygenase-2 overexpression and signs of chronic inflammation in colon mucosa but only after 13 weeks and not at 7 weeks of ω-6 rich diet administration [37]. The diets used in these two studies did not contain any measurable ARA amount but LA was the main ω-6 fatty acid. Regarding a specific influence of dietary ARA intake on the gut microbiome, Zhuang et al. [38] found that 1% ARA diet supplementation favored pro-inflammatory microbiota by reducing the *Firmicutes*/*Bacteroidetes* ratio and induced systemic and hypothalamic inflammation only in male C57BL/6j mice while it increased obesity for both genders. The particularity of this work is that the authors first induced obesity in mice by feeding them a high-fat diet without ARA (45% energy from fat or 23,5% lipid *w*/*w*) for 10 weeks, and then they tested the ARA impact on the gut microbiota by using a high-fat diet with or without ARA for an additional 15 weeks. Therefore, it is not possible to distinguish the ARA-specific influence from the obesity-associated effects. Furthermore, Benoit et al. showed that 20% lipid intake (*w*/*w*) altered gut microbiota composition and induced metabolic endotoxemia in mice, while a higher lipid intake (45% *w*/*w*) did not cause such effects [39]. Therefore, the lipid composition of the diet and its administration duration can noticeably modify pathophysiological effects of ARA on the gut–brain axis. In this paper, we investigated the effects of dietary intake of 1% ARA (1% *w*/*w*, corresponding to 417 mg/day for humans) in the context of a moderately hyperlipidic diet (15% fat *w*/*w*) compared with a similar diet without ARA and a conventional murine diet (5% fat *w*/*w*). The objectives were to determine whether a dietary ARA intake in the context of a moderately hyperlipidic diet, which could be consistent with those observed in human diets, could: (1) modify gut microbiota, (2) induce gut and systemic low-grade inflammation, and (3) facilitate the occurrence of neuroinflammation. We first studied the fecal microbiota composition and modifications in the expression levels of proteins involved in colon inflammation and permeability after 9 weeks of administration of the 3 diets. We further examined whether dietary ARA modified expression levels of pro-inflammatory cytokines in the liver and adipose tissue. Finally, we examined the expression of neuroinflammation markers as several studies suggested that microbiota modifications and low-grade gut inflammation could contribute to the occurrence of neuroinflammation and pathological features of Alzheimer’s disease [40,41].

## 2. Materials and Methods

### 2.1. Diet Design

Three diets were used in this study. The first diet, named the standard diet without ARA (Std-ARA) diet, was a conventional murine diet containing 5% lipids (*w*/*w*). The two other diets, named the “HL-ARA” diet and the “HL + ARA (1% ARA)” diet, contained 15% lipids (*w*/*w*). The composition of the 3 diets is given in Table 1. The Std-ARA diet provides less energy per weight of food than the two HL-ARA and HL + ARA diets (390 kCal/100 g of food instead of 430 kCal/100 g of food, respectively, Table 1).

Arasco oil (Life^TM^ ARA) was a gift from DSM Nutritional Product (Courbevoie, France), hemp oil was provided from Fermes d’Ormes (Ormes, France), and lard and other components were provided by UPAE (Unité de Préparation des Aliments Expérimentaux, INRAE, Jouy-En-Josas, France), which produced the 3 diets. Diet compositions were calculated from gas chromatography (GC) analyses of the various oils and lard, and checked after diet preparation. GC analyses were performed on a GC-2010 analyzer (Shimadzu) and a SM^®^-2380 capillary GC column (60 m × 0,25 mm, L × I.D., d_f_ 0.2 µm) by using C17 internal standard and double independent assays in LIBIO laboratory (Laboratory of Biomolecules Engineering, University of Lorraine). The lipid samples were analyzed using the BF3-methanol method [42]. Table 2 reports the fatty acid composition of each diet. The complete diet composition and lipid sources used in this study are given in the Appendix A. Diets were stored at 4 °C and protected from light to prevent oxidation.

### 2.2. Animal Handling

Male BALB/c mice were purchased from ENVIGO RMS SARL (Gannat, France) at 6 weeks of age and housed in the animal facilities for a 2-week adaptation period. During this period, the mice were fed the standard Teklad global 16% protein rodent diet (Envigo, Madison, WI, USA). Then, their baseline body weights were measured and fecal samples were collected for microbiota analysis (0 week). Mice were then randomly divided into three groups (15 animals in each group) and housed in individual cages (24 ×11 ×12 cm) to monitor food and water consumption. The three groups were respectively fed ad libitum with the Std-ARA diet (15% of lipids, without ARA supplementation, group N° 1), the HL-ARA diet (45% of lipids, without ARA supplementation, group N° 2) and the HL + ARA diet (45% of lipids with ARA supplementation, group N° 3). In our previous work, Thomas et al. [16] evidenced that dietary ARA intake worsened Aβ oligomers’ negative effects on learning abilities and expression levels of α-amino-3-hydroxy-5-methyl-4-isoxazolepropionic acid receptors (AMPA receptors) at 12 weeks of diet administration. Since, in the previous study, Aβ oligomers were injected into cerebral ventricles at 10 weeks after the beginning of the diet administration and behavior evaluations were performed during the last 2 weeks, we assumed that dietary ARA could induce effects before 10 weeks of administration. Therefore, we administrated our 3 diets to mice for 9 weeks. At the end of this period, fecal samples were collected before sacrifices. Blood, colon, liver, brain, and mesenteric adipose tissues were collected after sacrifices, rapidly frozen in liquid nitrogen, and then stored at −80 °C until biochemical analysis.

During the whole experiment, mice had access to food and water ad libitum. Body weights of mice were measured weekly. Hygrometric and thermal conditions were 50% ± 5% air humidity and 22 ± 2 °C, respectively. The light/dark cycle was also reversed (light off from 8:30 am to 8:30 pm) to facilitate the handling of the animals while respecting their biological cycle.

### 2.3. Fecal Microbiota Analysis

Total bacterial DNA was extracted from mice fecal samples according to Godon et al. [43]. Briefly, 250 µL of Guanidine isothiocyanate (4 M) (Sigma-Aldrich G9277, St. Louis, MI, USA) and 40 µL of laurylsarcosine (10%, Sigma-Aldrich L9150) were added to frozen feces (50 mg) samples. After 10 min thawing at room temperature, feces were resuspended in 500 µL of *N-*laurylsarcosine (5% in phosphate buffer pH 8, 0.1 M) and incubated at 70°C for 1 h with stirring. After grounding in a ball mill at 4 °C (Precellys^®^Evolution, Ozyme, France), a solution containing 1% polyvinylprrolidone (Supelco^®^ 77627) Tris-HCl 0,5 M was added to the grounded matrix and centrifuged at 4 °C for 5 min at 20,000× *g*. The pellet was resuspended in 1% polyvinylprrolidone Tris-HCl solution and centrifuged under similar conditions. The step of remaining pellet resuspension and centrifugation was repeated two more times under similar conditions. The supernatants obtained from all four extractions were pooled together and the DNAs were precipitated by adding an equal volume of isopropanol. After centrifugation for 10 min at 20,000× *g*, DNA pellets were resuspended in 450 µL PBS and 50 µL potassium acetate 3M (Sigma-Aldrich, P1190). After overnight incubation at 4 °C, the samples were centrifuged for 30 min at 20,000× *g* at 4 °C. The DNA-containing supernatants were incubated with 2 µL RNase A (Sigma-Aldrich, 10109142001) for 30 min at 37% under stirring. The DNAs were reprecipitated in the presence of 50 µL of sodium acetate and 3 M and 1 mL of ethanol 100%. After centrifugation for 10 min at 20,000× *g*, pellets were washed 3 times in ethanol 70% and then dried at room temperature for 2 h under hood. Dried pellets were resuspended in Tris HCL 10 mM, EDTA 1 mM. DNA concentrations were determined using the NanoDrop instrument (Ozyme, France). The V3–V4 hyper-variable region of the 16S rRNA gene was amplified with two primers: MSQ-16SV3F (5′-CTTTCCCTACACGACGCTCTTCCGATCTACGGRAGGCWGCAG-3′) and MSQ-16SV4R (5′-GGAGTTCAGACGTGTGCTCTTCCGATCTTACCAGGGTATCTAATCCT-3′). The polymerase chain reactions (PCRs) were carried out in a thermal cycler Mastercycler^®^ pro (Eppendorf, Hamburg, Germany) using mixtures containing 10 ng of bacterial DNA, 0.5 μM of primers MSQ16SV3F and MSQ-16SV4R, 0.2 mM of each dNTPs, and 0.5 of U DNA-free Taq-polymerase (MolTaq 16S DNA Polymerase, Molzym). Amplification started at 94 °C for 60 s, followed by 30 cycles of denaturation at 94 °C for 60 s, annealing at 65 °C for 60 s and extension at 72 °C for 60 s, and a final extension at 72 °C for 10 min. The resulting PCR products were purified and sent to the @BRIDGe platform (INRA, Jouy-en-Josas) for sequencing using Illumina MiSeq technology (Illumina, CA, USA).

Sequences were analyzed using the Galaxy-supported program FROGS to produce abundance tables of Operational Taxonomic Units (OTUs) and their taxonomic affiliation. The successive steps involved de-noising and clustering of the sequence into OTUs using SWARM; chimera removal using VSEARCH; and taxonomic affiliation for each OTU using the RDP Classifier on the SILVA SSU 123 database. Statistical analyses were performed using “R” language and environment version 3.2.3. Alpha- and beta-diversity measurements and analysis of the differences in OTUs between samples was performed using the add-on package “Phyloseq” [44]. The OTUs count table and taxonomic classification were subjected to pathway abundance analysis by Phylogenetic Investigation of Communities by Reconstruction of Unobserved States (PICRUSt2; [45]).

### 2.4. Immunoblotting Analysis

Cortex and hippocampus were homogenized by using mini-potters and pipette tips in 25 mM Tris pH 7.4, 150 mM NaCl, 1 mM EDTA, 1% (*v*/*v*) Nonidet NP-40, 1% (*m*/*v*) desoxycholate sodium, 0.1% (*m*/*v*) SDS, 1 mM PMSF, 1 mM Na3Vo4, and protease inhibitor cocktail « Complete » (Roche, France). After two freezing and thawing cycles, the samples were centrifuged for 30 min at 10,000 g and 4 °C to remove nuclei and cell debris. The supernatants were stored at −80 °C. Whereas, proteins from colon samples were extracted by a QIAGEN AllPrep DNA/RNA/Protein Mini Kit (Hilden, Germany) following the supplier’s recommendations. Proteins in all supernatants were quantified by BCA Protein Assay Kit (Thermo- Fischer Scientific, Waltham, MA, USA) before Western-blot analysis.

For Western-blot analysis, sample supernatants were mixed with equal volumes of 2× Laemmli buffer with β-mercaptoethanol 0.004% (*w*/*v*) and denatured by heating the mixture at 95 °C for 5 min. Then, samples were separated on 12% sodium dodecyl sulfate-polyacrylamide gel electrophoresis (SDS-PAGE) in a Mini-Protean II system (Biorad, USA) and transferred to nitrocellulose membranes (GE Healthcare). Blots were probed with mouse IgG anti-GFAP primary antibody (anti-GFAP 1:1000, Sigma-Aldrich), goat IgG anti-Iba1 (1:1000, Novusbio), rabbit IgG anti-Claudine-1 (1:2000, Sigma-Aldrich), or mouse IgG anti-beta-tubulin (1:2000, Sigma-Aldrich), followed by HRP-conjugated secondary anti-mouse IgG, anti-rabbit IgG (1:5000, Sigma-Aldrich), or anti-goat IgG (1:2000, Novus). Immuno-complex bands were detected using the enhanced chemiluminescence (ECL) protocol (GE Healthcare). The bands’ intensities were then quantified on a Chemidoc densitometer using Image Lab^TM^ software (Bio-rad Laboratories, Hercules, CA, USA).

### 2.5. RNA Extraction and RT-qPCR Analysis

Total RNAs were extracted from the colon and liver samples using QIAGEN AllPrep DNA/RNA/Protein Mini Kit (Hilden, Germany) and from the brain and adipose tissues using QIAGEN RNeasy Lipid Tissue Mini Kit (Hilden, Germany) according to manufacturer’s instructions. cDNAs were reverse transcripted using PrimeScript™ RT Master Mix (Takara Bio Inc., Shiga, Japan) according to the manufacturer’s protocol. The primers used for qPCR analysis are presented in Table 3.

Real-time PCRs were performed in duplicate using the Biorad CFX Real-Time PCR system (Bio-rad Laboratories, Hercules, CA, USA) and TB Green^®^ Premix Ex Taq™ (Tli RNase H Plus) (Takara Bio Inc., Shiga, Japan). Relative quantification was performed according to Pfaffl’s method [46] and expression was normalized against mRNA levels of GAPDH.

### 2.6. Immunohistochemical Analysis

Coronal slices of mouse hemispheres (12 μm) were fixed in 4% paraformaldehyde for 15 min and then washed thrice with phosphate buffer saline 1× pH 7.4 for 5 min. They were permeabilized for 10 min with 0.1% triton in 1×-PBS and incubated for 1 h at room temperature in blocking buffer (10% BSA, 0.1% triton in PBS (2,7 mM, 1×). Slices were then incubated overnight with primary antibody mouse IgG anti-GFAP (1:800, Dako) in blocking solution at 4 °C under stirring. After washing 3 times in 1x-PBS for 5 min, sections were incubated with secondary antibody goat anti-mouse IgG AlexaFluor 488 (1:1000, Invitrogen) diluted in blocking solution for 2 h at room temperature. Finally, slices were mounted with ProlongTM Gold antifade mounting media containing DAPI (4’,6-diamidino-2-phenylindole) to label cell nuclei (Thermo Fisher Scientific). 

After observation with a fluorescence microscope (Nikon, Nikon Instruments Europe B.V.), staining was quantified in four images from each brain area studied. The photographs were captured by NIS Element (Nikon) software and ImageJ software was used to count the cells and measure the surface area (mm^2^) of the brain structures. For the quantification of results, surface area, number of labeled positive cells, and thickness of each slice (12 μm) were used to calculate the density of positive cells (number of positively labeled cells per mm^2^) in the mouse brains (*n* = 2 per group).

### 2.7. Statistical Analysis

Results are presented as mean ± SEM. The statistical analysis was performed using one-way ANOVA followed by Bonferroni’s multiple comparison post hoc test or Kruskal–Wallis test followed by Dunn’s multiple comparisons test for post hoc analysis using GraphPad Prism Software (V9.4, San Diego, CA, USA). Regarding the analysis of fecal microbiota, ANOVA and DeSeq were performed. 

## 3. Results

### 3.1. Impact of Dietary ARA Intake on Murine Growth and Fecal Microbiota

After 9 weeks of administration of the 3 diets, i.e., Std-ARA, HL-ARA, and HL + ARA, no significant difference in food intake and mouse weight growth were observed in all the diet groups (Figure 1A,B). Furthermore, the effect of different diets on organ weights was also studied. The only difference that we observed among the three groups was the reduction in mesenteric adipose tissue weight in mice fed on HL + ARA diet (Figure 1C).

Fecal microbiota composition was studied at week 0 (initiation of the experiments) and week 9 to determine whether high-fat and ARA-rich diets induced variations of this microbiota. The 3 mice groups displayed a decreased α- and β-diversity between week 0 and week 9 of diet administration (Appendix A). At week 9, no significant difference in β-diversity among the three diet groups was observed (Appendix A). However, evaluation of phyla abundances at week 9 indicated that the HL-ARA diet induced an increase in *Actinobacteriota* phylum compared with the Std-ARA group and that ARA intake in the HL + ARA group suppressed it (Figure 2A,B). It is worth noting that wide variations exist within each mouse group even though significant differences have been highlighted among groups for *Actinobacteriota* phylum. Further investigations revealed that the modification of the *Actinobacteriota* phylum was due to the proliferation of the *Bifidobacterium pseudolongum* species which was favored by the hyperlipidic HL-ARA diet contrary to HL + ARA diet (Figure 2C). Unlike the *Bifidobacterium* genus, the *Escherichia–Shigella* genus in the *Proteobacteria* phylum was less abundant in mice fed the HL-ARA diet compared with those fed the Std-ARA diet, and the HL + ARA diet where dietary ARA intake favored the overgrowth of this genus (Figure 2D). Higher lipid intakes also favored the proliferation of the *Bilophila* and *Blautia* genera regardless of the absence or presence of ARA in these diets (Figure 2E,F).

Based on the fecal microbiota analysis of mice fed the Std-ARA, HL-ARA, and HL + ARA diets, the metabolic pathways present in the microbiota of the three mice groups were compared. The lowest number of significant differences was observed between the Std-ARA and HL + ARA groups (seven pathways with different abundances, Table 4). Comparison of Std-ARA and HL-ARA groups led to an intermediate number of pathways (20 metabolic pathways, Table 4). Remarkably, the highest number of differences was observed between the HL-ARA and HL + ARA groups, although the only difference between these two diets was the absence or presence of ARA. Among the 38 metabolic pathways identified in this last comparison (Table 4), 20 were less abundant in the HL + ARA group than in the HL-ARA group and 18 were more abundant. Among the most represented genes in the HL-ARA group, some are involved in amino-acid and bacterial wall synthesis, while the most represented in the HL + ARA group are involved in the synthesis of folates, flavins, and B1 and B6 synthesis (Table 4).

Regarding Std-ARA and HL-ARA groups, the less represented metabolic pathways in the HL-ARA group (13 metabolic pathways) are involved in amino acid (arginine, isoleucine, glutamine, and tryptophan), heme, and protein synthesis, whereas those less abundant in the Std-ARA group (7 pathways) play a role in quinone and nucleotide synthesis. Only four metabolic pathways were found more abundant in the Std-ARA group compared with the HL + ARA group and are involved in the reductive TCA cycle, tryptophan, and tetrapyrrole synthesis (Table 4), while three were less abundant and are involved in nucleotide synthesis for two of them and glycolysis for the last one (Table 4).

### 3.2. Impact of Dietary ARA Intake on the Expression Level of Inflammatory Markers in Colon

The *Escherichia–Shigella* genus is associated with intestinal inflammation [47], while *Bifidobacterium pseudolongum* exhibits anti-inflammatory activities [48], based on our results that dietary intake of ARA promoted the growth of the *Escherichia–Shigella* genus but reduced the growth of *Bifidobacterium pseudolongum*. The expression levels of pro-inflammatory cytokines were investigated in colon samples collected from the 3 groups of mice at 9 weeks of diet. Dietary ARA significantly increased IL-1β and CD40 expression levels by 3.8- and 15.1-fold, respectively, in the HL + ARA group compared with those observed in the Std-ARA group (Figure 3A,B). In contrast, no variation was observed for their expression in the HL-ARA group compared with the Std-ARA group (Figure 3A,B). While CD40/TNFRS5 is one of the TNF-α receptors involved in colitis associated to IL-1β production [49], no significant variation was found in TNFα, IL-6, and adiponectin expression levels between HL-ARA and HL + ARA groups compared with the mice fed the Std-ARA diet (Appendix A, Appendix A).

Furthermore, the expression levels of claudin-1 and occludin that form the tight junctions of the gut were examined to evaluate the putative influence of the various diets on colon permeability. We did not observe any modification of the expression levels of these genes in the three groups of mice (Appendix A). These data do not support the increase in gut permeability due to higher lipid or ARA intake.

### 3.3. Relationship between Low Grade Inflammation in Liver and Adipose Tissues and ARA Intake

To investigate whether ARA and higher lipid intake could induce systemic low-grade inflammation, the expression levels of pro-inflammatory cytokines IL-1β, IL-6 and TNF-α were measured in liver and mesenteric adipose tissue. The results showed no significant modification of IL-1β and IL-6 expression levels in the liver and in mesenteric adipose tissue (Appendix A). Regarding TNFα, its expression level in mesenteric tissue of mice fed the ARA-rich diet was drastically reduced by 31.6-fold compared with the mice fed the HL-ARA diet. However, the mice fed the HL-ARA- diet exhibited 3.4-fold higher expression levels on average in comparison with those observed in mice fed the Std-ARA diet without reaching a level of significance because of the dispersion of the values among individuals (Figure 4).

### 3.4. Impact of Dietary ARA on Brain Glial Cells and Pro-Inflammatory Cytokine Expression

Since dietary ARA intake favor proliferation of pro-inflammatory bacterial species and increased expressions of several low-grade inflammatory markers in gut, we investigated whether it could modulate the expression of pro-inflammatory cytokines and glial markers in the brain. Firstly, the expression of the pro-inflammatory cytokines IL-6 and IL-1β was studied in mouse half-brain samples obtained after 9 weeks of administration of the various diets. The results showed reductions in expression levels of IL-6 by 2.2- and 3.5-fold in mice fed the HL-ARA and HL + ARA diets, respectively, compared with those fed the Std-ARA diet (Figure 5A). Regarding IL-1β, we found an average reduction of 5.9-fold in brains of mice fed the HL-ARA diet compared with those fed the Std-ARA diet without reaching a significance level (Figure 5B). In contrast, dietary ARA intake increased by 1.8- and 10.8-fold in IL-1β expression levels compared with those found in mice fed the Std-ARA and HL-ARA diets, respectively (Figure 5B). Therefore, we further investigated the expression levels of the CD40 receptor and the CD40 responsive cytokine IL-12, both of which are more expressed in microglial cells [50]. CD40 mRNA levels were decreased by 2.1- and 2.6-fold on average, in the brain of HL-ARA- and HL + ARA-fed mice, respectively, compared with Std-ARA-fed mice, without reaching a level of significance (Figure 5C). In contrast, those of IL-12 were drastically reduced by 50- and 13.3-fold by HL-ARA and HL + ARA diets, respectively, compared with the control Std-ARA mice group (Figure 5D).

Furthermore, we examined if the lipid intake influenced the expression of GFAP and microglial markers in the cortex and hippocampus and whether they could be correlated to lower expression of IL-12. The protein expression levels of the microglial marker Iba1 were determined in the cortex and hippocampus of the mice of the three diet groups. The results revealed that HL-ARA and HL + ARA diets did not display any variations in expression levels of Iba1 in the cortex and hippocampus compared with those measured in the same brain structures of Std-ARA-fed mice (Figure 6A–C). In contrast, the protein expression of the astrocyte marker GFAP was increased by 1.4- and 1.8-fold in cortex and hippocampus, respectively, of HL + ARA-fed mice compared with the two other mice groups (Figure 6B,D).

The increased expression of GFAP in the hippocampus of the HL + ARA diet group compared with the Std-ARA and HL-ARA mice groups was confirmed by immunohistochemistry results (Figure 7).

## 4. Discussion

The influence of dietary ARA intake on microbiota and low-grade inflammation through the gut–brain axis was investigated in this paper. ARA was provided as a component of a moderately hyperlipidic diet in which the fat represented 30% of energy source and the ω-6/ω-3 ratio was in the range of 5–6, which is the recommended value of the French agency for food safety, AFSSA. The fatty acid profiles of the three diets in terms of percentage of each fatty acid were almost similar except the HL + ARA diet that contained 6.6% ARA and 25.3% LA instead of 31.9% LA for the HL-ARA diet. The HL + ARA diet contained slightly less ALA acid while its ω-6/ω-3 ratio remained in the recommended value (Table 2). The influence of the ARA-containing diet (HL + ARA) on the gut microbiota of mice was compared with those of the mice fed the standard diet containing 5% fat without ARA (Std-ARA) and a hyperlipidic diet without ARA (HL-ARA), with both of them displaying a similar ω-6/ω-3 ratio and the same fatty acid profiles.

Concerning microbiota analysis, the first remarkable observation was the proliferation of *Bifidobacterium pseudolongum* in the microbiota of mice fed the HL-ARA diet. Indeed, *Bifidobacterium pseudolongum* was involved in triglyceridemia reduction in mice fed a high-fat diet [51] and displayed anti-inflammatory activity [52]. In addition, *Bifidobacteria* strains are reported to convert linoleic and linolenic acids into conjugated derivatives which have potentially positive effects on well-being [53]. Therefore, the proliferation of *Bifidobacterium pseudolongum* can result from high triglyceride intake containing a majority of mono- and polyunsaturated fatty acids with an optimal ω-6/ω-3 ratio such as in the HL-ARA diet. Paradoxically, the HL + ARA diet suppressed *Bifidobacterium pseudolongum* growth and favored the proliferation of *Escherichia–Shigella* genus. The latter are anaerobic facultative symbionts that can promote gut inflammation [54,55] and are involved in several human pathologies [55]. Their proliferation is reduced by using probiotic *Bifidobacteria* strains [56]. However, how ARA differentially modulate the growth of *Bifidobacterium pseudolongum* and the *Escherichia–Shigella* genus is questionable, and several hypotheses can be made. First, it is reported that polyunsaturated fatty acids such as LA (present in high amounts in HL-ARA and HL + ARA diets) and ARA have antibacterial activities [57] but their respective specificity relative to a bacterial species such as *Bifidobacterium pseudolongum* is still unknown. Second, eicosanoids, which result from ARA conversion, could influence microbiota. Indeed, ARA released by pancreatic lipase in the small intestine can be incorporated into immuno-competent cells and be converted into various eicosanoids. Adam et al. [20] observed an increase in ARA-containing lipids and production of eicosanoids, mainly 5- and 12-hydroxyeicosatetraenoic acids (5- and 12-HETE), in juvenile *Danio rerio* fish fed a 2% ARA diet compared with a 0.19% ARA diet. Naoe et al. [58] measured high amounts of prostaglandins and leukotrienes in the spleen and small intestine among 9 tissues and plasma of mice fed a 1% ARA-supplemented diet. Prostaglandins and leukotrienes can stimulate the activities of polymorphonuclear cells and macrophages, which can consecutively influence gut microbiota composition [57]. However, the precise mechanism by which ARA alters gut microbiota and specifically *Bifidobacterium pseudolongum* requires further studies. Moreover, another microbiota modification with both hyperlipidic diets, i.e., HL-ARA and HL + ARA in comparison with the Std-ARA diet, was the higher proliferation of the *Blautia* and *Bilophila* genera. *Blautia* proliferation is associated with high fat diets in mice [59] and with the consumption of processed foods and animal-derived foods in humans [60]. However, the consequences of the *Blautia* proliferation are questionable since it has been correlated with endotoxin synthesis [50] but also with anti-obesogenic effects [61]. Proliferation of the bile-resistant *Bilophila* genus is also favored by high-fat diets [62]. Some species such as *Bilophila wadsworthia* have a pejorative impact on metabolic regulations [63].

Gut microbiota strongly influence the well-being of the host. This importance is based on the possible complementation of the host metabolic pathways by those of the microbiota, such as synthesis of precursors and vitamins which are not synthesized by mammals. Therefore, metabolic pathways associated with the microbiota of the three mice groups were examined. In silico study on metabolic pathways highlighted a greater number of putative differences between the two hyperlipidic diets as well as between the Std-ARA diet and the two previous ones, while they only differ by the presence or absence of ARA and not by their lipid contents. Dietary ARA addition might reduce several metabolic pathways such as complex carbohydrate synthesis, bacterial wall building, essential amino acids synthesis, and increase several other ones such as vitamin synthesis, precursors synthesis, and degradation of complex carbohydrates. It should be noted that this in silico study did not indicate the pathways involved in the synthesis or degradation of lipids. These hypotheses require to be confirmed by experimental works such as Chakrabarti’s study that investigated the influence of the *Escherichia coli* M8 strain on lipid metabolism in gnotobiotic mice [64].

Downstream of the influence of dietary ARA intake on gut microbiota, its impact on inflammation levels in gut tissue was studied. Although some studies indicated that dietary ARA does not increase the severity of colitis induced by exogenous agents such as dextran sulfate sodium [65,66], the results of this work showed that it led to higher expression levels of the pro-inflammatory cytokine IL-1β and CD40 protein in colon in the context of a moderately hyperlipidic diet. The stimulations of their expressions were in the range of 4–15-fold, which is consistent with a low-grade state of inflammation and not a high-grade inflammation such as observed in models of Crohn’s disease or ulcerative colitis. The increased expression levels of IL-1β are consistent with its previously described role in metabolic inflammation in response to the NLRP3 inflammasome [67]. Furthermore, CD40 is a receptor, along with its ligand CD40-L, involved in the fine regulation of immune cell interactions and inflammation regulation, including that associated with nutrients and insulin response [68]. Overexpression of CD40 is observed in the colonic mucosa of patients affected by Crohn’s disease and ulcerative colitis [69,70]. Moreover, a constitutive expression of CD40 in dendritic cells resulted in the occurrence of fatal colitis in transgenic mice [47]. The gut microbiota could influence the stimulation of CD40 [71]. In particular, *Bifidobacterium longum* alleviated dextran-sulfate-induced colitis through CD40- and CD80-mediated IL-12 production in intestinal epithelial cells [72]. However, CD40-CD40-L dyads can also contribute to the anti-inflammatory IL-10 production in B-cell interactions with mast cells [73].

Numerous works reported that diet-induced microbiota dysbiosis and gut inflammation are associated to low-grade inflammation in liver and adipose tissue [74]. Therefore, expression levels of pro-inflammatory cytokines were measured in these two latter tissues despite the fact that we did not obtain evidence supporting the modification of the gut barrier integrity. We did not evidence any variation of expression levels of pro-inflammatory cytokines among the three diet-groups in liver. In mesenteric adipose tissue, there is a trend towards an increase in the TNFα expression levels in mice fed the HL-ARA diet compared with those fed the Std-ARA diet. Dietary ARA intake significantly reduced this TNFα expression level (Figure 4). The same trend was observed for IL-6 without reaching a significant level (Appendix A). Contrary to Zhuang et al. [38], we did not observe obesity in mice fed both the hyperlipidic HL-ARA or HL + ARA diets. Zhuang et al. [38] first induced obesity by feeding C57BL/6J male and female mice with a 45% fat diet for 10 weeks before supplementing with ARA for an additional 15 weeks in a mice group. Furthermore, the authors used fat milk as a lipid source, which usually contains more saturated than unsaturated fatty acids. Substituting mono- or polyunsaturated to saturated fatty acids in high-fat diets reduced the increase in body weight and adiposity in rodents [75]. In Zhuang et al.’s study [38], ARA enhanced obesity in female and male mice but favored pro-inflammatory microbiota, promoted browning of adipose tissue, and increased systemic and hypothalamic inflammation levels only in male mice. In our study, we did not induce obesity after a feeding period shorter than that used in Zhuang et al. [38], but a reduction in mesenteric adipose tissue weight was observed which was more marked in mice fed the HL + ARA diet. Wernstedt Asterholm et al. [76] showed that inadequate inflammatory response in adipose tissues through suppression of response to TNFα is associated with reduced adipogenesis, lack of weight, and leaky gut. Therefore, ARA dietary intake might favor inadequate adipose tissue growth and metabolic response.

In a last step, the impact of dietary ARA intake on the brain was examined. We previously demonstrated that dietary ARA altered learning abilities and expression of AMPA receptors in mice after intracerebroventricular injection of Aβ_42_ [16]. On the basis of this previous work and the above present data on gut microbiota and higher expression levels of pro-inflammatory cytokines in colon, evidence of neuroinflammation could be expected. Indeed, moderate GFAP overexpression was observed in the hippocampus and cortex of mice fed the HL + ARA diet for 9 weeks, compared with mice fed the Std-ARA and HL-ARA diets. However, downregulation of IL-6 and IL-12 expression associated with a lack of significant difference of expression of the microglial Iba1 marker was also observed. The mRNA levels of Il-1β, IL-6, CD40, and IL-12 were measured in extracts of half-brains. The results represented an average of the levels in the different brain structures, which may be higher in some regions such as the hippocampus. Furthermore, IL-6 and IL-12 are not only involved in inflammation but also in several neuronal functions. IL-6 stimulates neurogenesis in the hippocampus as well in the hypothalamus, in which its activity on neuronal cells contributes to restoring food control and energy expenditure [77]. It has also been reported that IL-12 promotes neurite growth [78]. Besides its activity on neuronal cells, reduced expression of IL-12 could alter brain defense against neurotoxic agents such as Aβ peptide oligomers since inflammation has a dual role, protective and destructive, in Alzheimer’s disease. Genetic deletion of the p40 subunit of IL-12 differentially affects the Aβ burden or production in transgenic male and female mice which overproduce the Aβ peptide [79]. We assumed that the higher sensitivity of the mice fed the ARA-rich diet to the neurotoxicity of Aβ peptide oligomers could be due to a neuronal weakening or a disturbance of the glial cells to degrade these oligomers before the intracerebroventricular Aβ injections. Additional works are necessary to confirm this hypothesis, which constitute the first weakness of this work. For example, extensive investigations on the putative involvement of several actors of the gut–brain axis such as short-chain fatty acids or other microbial metabolites, adipokines, or other cytokine family members in the various organs may provide responses. The second weakness is that this study was performed in mice and cannot be extrapolated to humans without caution. However, we have shown for the first time that dietary ARA impacts the brain following microbiota modifications and induction of low-grade inflammation in the colon in the context of a moderately hyperlipidic diet in mice. In addition, we have shown that an increase in dietary lipid intake from 5% to 15% with a high amount of unsaturated fatty acids favors the proliferation of the anti-inflammatory *Bifidobacterium pseudolongum* species in murine gut microbiota. These two findings constitute the main strengths of this work.

Many human foods contain ARA such as red meat, chicken, eggs, or fish, but ARA is also present in processed foods in a hidden form. A recent evaluation in 178 countries estimated that daily ARA intake varies from 100 to 350 mg/day and contributes <0.1% of total daily energy [6]. Based on these evaluations, the authors reviewed the studies and established a correlation between ARA intake and human health outcomes [80]. However, they focused their review on the incorporation levels of ARA in the various organs or tissues. To the best of our knowledge, there is no study until now on the influence of dietary ARA intake and gut microbiota modification. It would be of interest to investigate these effects in humans and to explore ARA influence on bacterial metabolism and gut physiology.

## 5. Conclusions

Overall, the present study characterizing the dietary arachidonic impact in the context of a moderately hyper lipidic diet from gut microbiota to brain through colon, liver, and adipose tissues, showed that the increase in lipid intake from 5% to 15% favors the proliferation of the anti-inflammatory *Bifidobacterium pseudolongum* in gut microbiota at 9 weeks of diet administration. However, 1% ARA intake favored the pro-inflammatory *Escherichia–Shigella* genus, led to low-grade colic inflammation, and induced astrogliosis in the brain. These changes in gut microbiota and systemic low-grade inflammation induced by dietary arachidonic acid may further promote the development of neurodegenerative pathologies. The involved mechanisms remain to be investigated.

## Figures and Tables

**Figure 1 nutrients-14-05338-f001:**
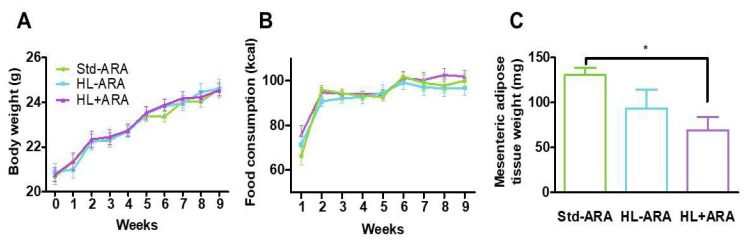
Kinetics of animal body weight (**A**) and food consumption (**B**) in male BALB/C mice fed on standard diet (Std-ARA) or lipid-rich diets without arachidonic acid (HL-ARA) or with (HL + ARA) for 9 weeks. Mesenteric adipose tissue weight (**C**) at the end of the study. Data are represented as the mean ± standard error of the mean (SEM). *n* = 15 per diet group for body weight and food consumption and *n* = 5 for mesenteric adipose tissue weight. * *p* < 0.05 compared with the standard diet, data analyzed by one-way ANOVA test and Bonferroni’s multiple comparisons test for post hoc analysis.

**Figure 2 nutrients-14-05338-f002:**
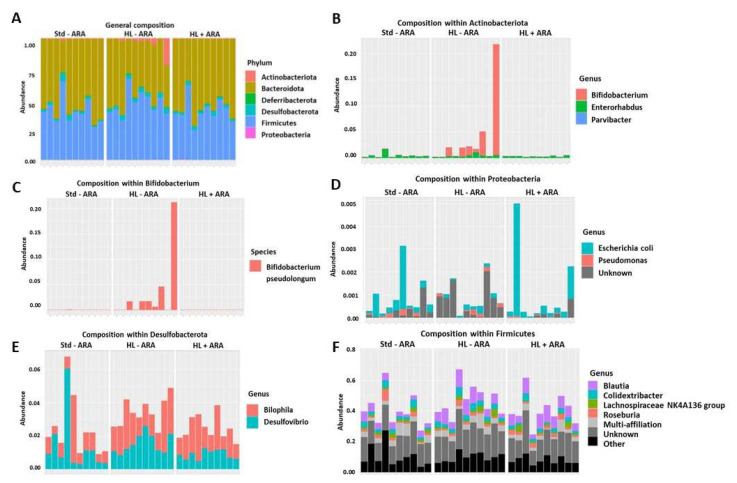
Fecal phylum and genus compositions of mice fed on the Std-ARA, HL-ARA, and HL + ARA diets for 9 weeks. The fecal microbiota compositions were determined on 10 mice in each group by using the V3–V4 hyper-variable region of the 16S rRNA gene. The relative abundance of the various phyla is shown in bar plot (**A**). Bar plots show the relative abundance of the various genera in the *Actinobacteria* (**B**), the *Proteobacteria* (**D**), the *Desulfobacteria* (**E**), and the *Firmicutes* (**F**) phyla. The species composition of the *Bifidobacterium* genus is shown in bar plot (**C**).

**Figure 3 nutrients-14-05338-f003:**
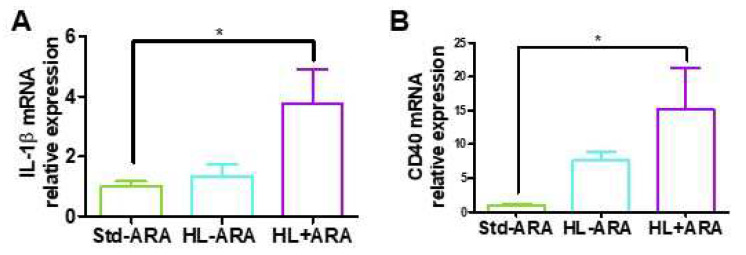
Colon gene expression of IL-1β (**A**) and CD40 (**B**) at 9 weeks of diet. Data are represented as the mean ± standard error of the mean (SEM). *n* = 4–5 mouse per diet group. * *p* < 0.05 compared with the standard diet. Data analyzed by either one-way ANOVA test and Bonferroni’s multiple comparisons test for post hoc analysis or Kruskal–Wallis test and Dunn’s multiple comparisons test for post hoc analysis.

**Figure 4 nutrients-14-05338-f004:**
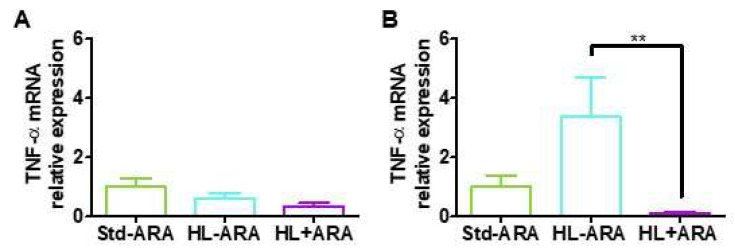
Gene expression of TNF-α in the liver (**A**) and mesenteric adipose tissue (**B**) at 9 weeks of diet. Data are represented as the mean ± standard error of the mean (SEM). *n* = 4–5 mouse per diet group. ** *p* < 0.01 compared with the standard diet. Data analyzed by one-way ANOVA test and Bonferroni’s multiple comparisons test for post hoc analysis.

**Figure 5 nutrients-14-05338-f005:**
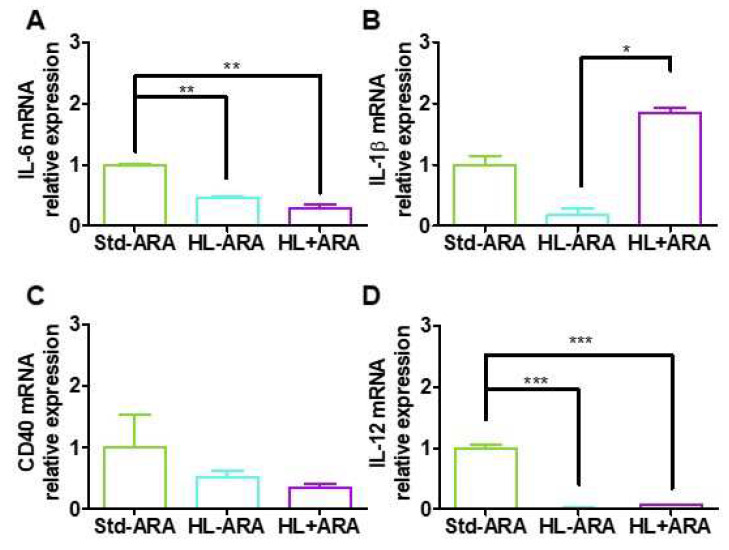
Half-brain gene expression of IL-6 (**A**), IL-1β (**B**), CD40 (**C**), and IL-12 (**D**) at 9 weeks of diet. Data are represented as the mean ± standard error of the mean (SEM). *n* = 2 mouse per diet group. * *p* < 0.05, ** *p* < 0.01, and *** *p* < 0.001. Data analyzed by one-way ANOVA test and Bonferroni’s multiple comparisons test for post hoc analysis or Kruskal–Wallis-test and Dunn’s multiple comparisons test for post hoc analysis.

**Figure 6 nutrients-14-05338-f006:**
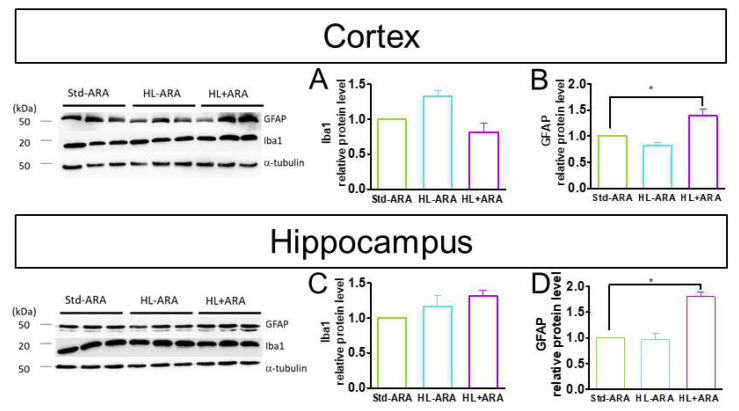
Western blot analysis of Iba1, GFAP, and α-tubulin in the cortex (**A**,**B**) and hippocampus (**C**,**D**). The protein levels were quantified by densitometry, normalized to the α-tubulin level, and expressed as a relative protein level. Data are represented as the mean ± standard error of the mean (SEM). *n* = 3 mouse per diet group. * *p* < 0.05 compared with the standard diet. Data analyzed by one-way ANOVA test and Bonferroni’s multiple comparisons test for post hoc analysis.

**Figure 7 nutrients-14-05338-f007:**
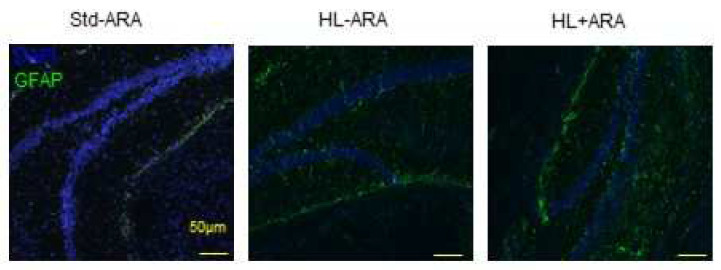
GFAP immunostaining (in green) in the hippocampus of mice fed the standard diet (Std-ARA) or hyperlipidic diets without arachidonic acid (HL-ARA) or with (HL + ARA) for 9 weeks. Nuclei were stained with DAPI (4’,6-diamidino-2-phenylindole, blue staining).

**Table 1 nutrients-14-05338-t001:** Nutritional profile of experimental diets.

	Std-ARA	HL-ARA	HL + ARA
Proteins (% *w*/*w*)	14	14	14
Carbohydrates (% *w*/*w*)	71	61	61
Lipids (% *w*/*w*)	5	15	15
Others (% *w*/*w*)	10	10	10
Energy (kcal/100 g) ^1^	390	437	437

^1^ Calculated according to Atwater system.

**Table 2 nutrients-14-05338-t002:** Fatty acid composition (% *w*/*w*) of experimental diets.

Fatty Acids	Std-ARA	HL-ARA	HL + ARA
C10:0 to C15:0	1.00	0.96	1.25
C16:0 palmitic acid	17.25	17.11	17.50
C16:1 palmitoleic acid	1.54	1.58	1.48
C17:0 3-hydroxyheptadecanoic acid	0.15	0.14	0.18
C18:0 stearic acid	8.60	8.63	9.16
C18:1 ω-9 oleic acid	31.84	31.65	32.07
C18:2 ω-6 linoleic acid	29.83	29.87	23.23
C18:3 ω-6 γ-linolenic acid	1.11	1.12	1.18
C18:3 ω-3 α-linolenic acid	7.42	7.48	5.33
C18:4 ω-3 stearidonic acid	0.57	0.54	0.52
C20:1 ω-9 11-eicosenoic acid	0.31	0.31	0.41
C20:4 ω-6 arachidonic acid	0.21	0.25	6.99
Ʃ saturated fatty acids	27	26.8	27.1
Ʃ mono-unsaturated fatty acids	33.7	33.5	34
Ʃ ω-6 PUFA	31.2	31.3	31.4
Ʃ ω-3 PUFA	8	8	5.9
ω-6/ω-3 PUFA ratio	3.9	3.9	5.3

**Table 3 nutrients-14-05338-t003:** Primer sequences used for RT-qPCR.

Gene	Primer Sequence (5’–3’)	
GAPDH	Forward:	AATTCAACGGCACAGTCAAGGC
	Reverse:	CGTGGTTCACACCCATCACAAA
IL-1β	Forward:	AGGCCACAGGTATTTTGTCGT
	Reverse:	TGTCCAGATGAGAGCATCCAG
IL-6	Forward:	TAGTCCTTCCTACCCCAATTTCC
	Reverse:	TTGGTCCTTAGCCACTCCTTC
TNF-α	Forward:	AGCCCACGTCGTAGCAAAC
	Reverse:	GATAGCAAATCGGCTGACGG
IL-12	Forward:	ACCTGTGACACGCCTGAAG
	Reverse:	CTCAGAGTCTCGCCTCCTTTG
Adiponectin	Forward:	GACGACACCAAAAGGGCTCA
	Reverse:	AGGTGAAGAGAACGGCCTTG
CD40	Forward:	TTGTTGACAGCGGTCCATCT
	Reverse:	TCACGACAGGAATGACCAGC
Claudin 1	Forward:	GCCATCTACGAGGGACTGTG
	Reverse:	CACTAATGTCGCCAGACCTGAA
Occludin	Forward:	TGAATGGGTCACCGAGGGAG
	Reverse:	AGATAAGCGAACCTGCCGA

GAPDH, glyceraldehyde-3-phosphate dehydrogenase; IL-1β, interleukin-1 beta; IL-6, interleukin-6; TNF-α, tumor necrosis factor-alpha; IL-12, interleukin-12; and CD40, cluster of differentiation 40.

**Table 4 nutrients-14-05338-t004:** Significant differences in metabolic pathways from Kyoto Encyclopedia of Genes and Genomes (KEGG) analysis of microbiota in mice fed the Std-ARA, HL-ARA, and HL + ARA diets. The type of comparison between the three groups of diet are indicated in bold in the first line and inside the table.

Comparison between HL-ARA and HL + ARA Groups				
Pathways	Description	Results	*p*-Values (Corrected)	Difference between Means
PWY-5676	Acetyl-CoA fermentation to butanoate, organic acid synthesis, energetic metabolism	HL-ARA > HL + ARA	0.0096	−0.0933
PWY-6471	Peptidoglycan biosynthesis IV (*Enterococcus faecium*), bacterial wall synthesis	HL-ARA > HL + ARA	0.0148	−0.0778
P124-PWY	Bifidobacterium shunt, degradation of complex carbohydrates, organic acid synthesis	HL-ARA > HL + ARA	0.0245	−0.0726
UDPNAGSYN-PWY	UDP-*N-*acetyl-D-glucosamine biosynthesis I, bacterial wall synthesis	HL-ARA > HL + ARA	0.0443	−0.0693
P4-PWY	Superpathway of L-lysine, L-threonine and L-methionine biosynthesis I	HL-ARA > HL + ARA	0.0062	−0.0690
P441-PWY	Superpathway of *N-*acetylneuraminate degradation, degradation of complex carbohydrates, organic acid synthesis	HL-ARA > HL + ARA	0.0171	−0.0652
PWY-5347	Superpathway of L-methionine biosynthesis (trans-sulfuration)	HL-ARA > HL + ARA	0.0252	−0.0649
MET-SAM-PWY	Superpathway of S-adenosyl-L-methionine biosynthesis	HL-ARA > HL + ARA	0.0349	−0.0528
OANTIGEN-PWY	O-antigen building blocks biosynthesis (*E. coli*), bacterial wall synthesis	HL-ARA > HL + ARA	0.0458	−0.0421
HOMOSER-METSYN-PWY	L-methionine biosynthesis I	HL-ARA > HL + ARA	0.0403	−0.0379
PWY-5304	Superpathway of sulfur oxidation (*Acidianus ambivalens*)	HL-ARA > HL + ARA	0.0401	−0.0349
PWY-7013	L-1,2-propanediol degradation	HL-ARA > HL + ARA	0.0115	−0.0338
PWY0-1479	tRNA processing, protein synthesis	HL-ARA > HL + ARA	0.0203	−0.0323
PWY-181	Photorespiration	HL-ARA > HL + ARA	0.0053	−0.0293
PWY-5005	Biotin biosynthesis II	HL-ARA > HL + ARA	0.0252	−0.0110
P341-PWY	Glycolysis V (Pyrococcus), anaerobic carbohydrate metabolism	HL-ARA > HL + ARA	0.0475	−0.0042
LEU-DEG2-PWY	L-leucine degradation I	HL-ARA > HL + ARA	0.0153	−0.0032
PWY-5741	Ethylmalonyl-CoA pathway	HL-ARA > HL + ARA	0.0263	−0.0010
PWY-5747	2-methylcitrate cycle II	HL-ARA > HL + ARA	0.0476	−0.0001
PWY0-42	2-methylcitrate cycle I	HL-ARA > HL + ARA	0.0492	−0.0001
PWY-4984	Urea cycle	HL-ARA < HL + ARA	0.0488	0.0111
HEME-BIOSYNTHESIS-II	Heme biosynthesis I (aerobic)	HL-ARA < HL + ARA	0.0459	0.0201
GALACTUROCAT-PWY	D-galacturonate degradation I, carbohydrate metabolism	HL-ARA < HL + ARA	0.0429	0.0226
RHAMCAT-PWY	L-rhamnose degradation I, carbohydrate metabolism	HL-ARA < HL + ARA	0.0120	0.0239
PWY-7323	Superpathway of GDP-mannose-derived O-antigen building blocks biosynthesis, bacterial wall synthesis	HL-ARA < HL + ARA	0.0449	0.0320
PANTOSYN-PWY	Pantothenate and coenzyme A biosynthesis I	HL-ARA < HL + ARA	0.0314	0.0389
PWY-6897	Thiamin salvage II, vitamin B1 synthesis	HL-ARA < HL + ARA	0.0149	0.0398
PWY0-845	Superpathway of pyridoxal 5’-phosphate biosynthesis and salvage, vitamin B6 synthesis	HL-ARA < HL + ARA	0.0354	0.0414
PWY-6892	Thiazole biosynthesis I (*E. coli*), vitamin B1 synthesis	HL-ARA < HL + ARA	0.0128	0.0423
PWY-6507	4-deoxy-L-threo-hex-4-enopyranuronate degradation, complex carbohydrate degradation	HL-ARA < HL + ARA	0.0437	0.0437
PYRIDOXSYN-PWY	Pyridoxal 5’-phosphate biosynthesis I, vitamin B6 synthesis	HL-ARA < HL + ARA	0.0341	0.0472
PYRIDNUCSYN-PWY	NAD biosynthesis I (from aspartate)	HL-ARA < HL + ARA	0.0395	0.0476
PANTO-PWY	Phosphopantothenate biosynthesis I	HL-ARA < HL + ARA	0.0324	0.0511
PWY-7539	6-hydroxymethyl-dihydropterin diphosphate biosynthesis III (Chlamydia), folate synthesis	HL-ARA < HL + ARA	0.0480	0.0534
RIBOSYN2-PWY	Flavin biosynthesis I (bacteria and plants)	HL-ARA < HL + ARA	0.0203	0.0537
THISYN-PWY	Superpathway of thiamin diphosphate biosynthesis I, vitamin B1 synthesis	HL-ARA < HL + ARA	0.0038	0.0592
FOLSYN-PWY	Superpathway of tetrahydrofolate biosynthesis and salvage, folate synthesis	HL-ARA < HL + ARA	0.0092	0.0669
PWY-6612	Superpathway of tetrahydrofolate biosynthesis, folate synthesis	HL-ARA < HL + ARA	0.0071	0.0692
**Comparison between Std-ARA and HL-ARA groups**				
ARGSYNBSUB-PWY	L-arginine biosynthesis II (acetyl cycle)	Std-ARA > HL-ARA	0.0071	−0.0887
PWY-5505	L-glutamate and L-glutamine biosynthesis	Std-ARA > HL-ARA	0.0155	−0.0809
PWY4FS-7	Phosphatidylglycerol biosynthesis I (plastidic)	Std-ARA > HL-ARA	0.0426	−0.0663
PWY4FS-8	Phosphatidylglycerol biosynthesis II (non-plastidic)	Std-ARA > HL-ARA	0.0426	−0.0663
PWY-7315	dTDP-*N-*acetylthomosamine biosynthesis, bacterial wall synthesis	Std-ARA > HL-ARA	0.0021	−0.0566
ARGSYN-PWY	L-arginine biosynthesis I (via L-ornithine)	Std-ARA > HL-ARA	0.0121	−0.0558
PWY-7400	L-arginine biosynthesis IV (archaebacteria)	Std-ARA > HL-ARA	0.0117	−0.0552
OANTIGEN-PWY	O-antigen building blocks biosynthesis (*E. coli*), bacterial wall synthesis	Std-ARA > HL-ARA	0.0196	−0.0547
PWY-5188	Tetrapyrrole biosynthesis I (from glutamate), heme synthesis	Std-ARA > HL-ARA	0.0029	−0.0536
PWY-5189	Tetrapyrrole biosynthesis II (from glycine), heme synthesis	Std-ARA > HL-ARA	0.0036	−0.0500
TRPSYN-PWY	L-tryptophan biosynthesis	Std-ARA > HL-ARA	0.0126	−0.0484
PWY-5104	L-isoleucine biosynthesis IV	Std-ARA > HL-ARA	0.0401	−0.0399
PWY0-1479	tRNA processing, protein synthesis	Std-ARA > HL-ARA	0.0338	−0.0368
PWY-7371	1,4-dihydroxy-6-naphthoate biosynthesis II, metaquinone synthesis	Std-ARA < HL-ARA	0.0411	0.0423
PWY-5659	GDP-mannose biosynthesis	Std-ARA < HL-ARA	0.0314	0.0524
PANTOSYN-PWY	Pantothenate and coenzyme A biosynthesis I	Std-ARA < HL-ARA	0.0375	0.0526
FASYN-ELONG-PWY	Fatty acid elongation, saturated lipid synthesis	Std-ARA < HL-ARA	0.0362	0.0654
PWY-7199	Pyrimidine deoxyribonucleosides salvage, nucleotide synthesis	Std-ARA < HL-ARA	0.0047	0.0701
PWY-5695	Urate biosynthesis/inosine 5’-phosphate degradation	Std-ARA < HL-ARA	0.0320	0.0743
PWY-6700	Queuosine biosynthesis, RNA synthesis	Std-ARA < HL-ARA	0.0295	0.0983
**Comparison between Std-ARA and HL + ARA groups**				
P23-PWY	Reductive TCA cycle I, lipids, carbohydrates, protein synthesis	Std-ARA > HL + ARA	0.0292	−0.0618
TRPSYN-PWY	L-tryptophan biosynthesis	Std-ARA > HL + ARA	0.0081	−0.0510
PWY-5188	Retrapyrrole biosynthesis I (from glutamate), heme synthesis	Std-ARA > HL + ARA	0.0142	−0.0440
PWY-5189	Tetrapyrrole biosynthesis II (from glycine), heme synthesis	Std-ARA > HL + ARA	0.0137	−0.0428
PWY-6121	5-aminoimidazole ribonucleotide biosynthesis I	Std-ARA < HL + ARA	0.0334	0.0155
GLYCOLYSIS	Glycolysis I (from glucose 6-phosphate)	Std-ARA < HL + ARA	0.0325	0.0391
PWY-7234	Inosine-5’-phosphate biosynthesis III, guanosine nucleotide synthesis	Std-ARA < HL + ARA	0.0279	0.0694

## Data Availability

Not applicable.

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
