# Peer review of "Impact of Dietary Arachidonic Acid on Gut Microbiota Composition and Gut–Brain Axis in Male BALB/C Mice"

_nutrients, 2022, doi:10.3390/nu14245338_

Round 1
Reviewer 1 Report
The connection of gut microbiota and brain activity is an actual and promising reseach direction. The obtained results have absolute practical application.
The one recommendation, addressed to the authors, is to strengthen the introduction by updating the data about the connection between brain activity and gut microbiota.
In general, the article may be recommended for the next step.
Author Response
The one recommendation, addressed to the authors, is to strengthen the introduction by updating the data about the connection between brain activity and gut microbiota.
Although it is difficult to make a short synthesis of the data about gut-brain axis due to the tremendous number of works devoted to this field, we tried to reinforce and update the data about it according to the reviewer’s recommendation by adding the sentences between lines 79 and 97 of the revised version. This includes 9 additional references which were published in 2022.
Reviewer 2 Report
Dear authors, the excellent manuscript that will contribute to current knowledge about arachidonic acid, I only make a few suggestions to improve the manuscript: “The higher consumption of ARA-rich meat and LA-rich oils such as sunflower and soybean oils led to the currently observed disequilibrium of the ω-6/ω-3 ratio in our societies”. 1. This phrase must be justified very well because moderate consumption of preparations such as salad dressing does not alter the relationship, perhaps a high consumption of ultra-processed foods or fried foods if they can be altered by a large amount of oil used in those preparations. 2. Missing to add the objective of the study at the end of the introduction 3. Statistics: a. Was a sample calculated? b. Was the normality of the data analyzed? 4. Discussion: I suggest adding weaknesses and strengths of the study, I also think a conclusion is necessary with what was mainly found
Author Response
“The higher consumption of ARA-rich meat and LA-rich oils such as sunflower and soybean oils led to the currently observed disequilibrium of the ω-6/ω-3 ratio in our societies”. 1. This phrase must be justified very well because moderate consumption of preparations such as salad dressing does not alter the relationship, perhaps a high consumption of ultra-processed foods or fried foods if they can be altered by a large amount of oil used in those preparations.
We deleted the sentence “The higher consumption of ARA-rich meat and LA-rich oils such as sunflower and soybean oils led to the currently observed disequilibrium of the ω-6/ω-3 ratio in our societies” and we replaces it by the sentence: “Regarding the dietary intakes of ARA and DHA, international evaluations showed that there are large variations worldwide. Many countries do not meet the nutritional recommendations [6]” (lines 41-43 of this revised version). We refer in this sentence to an international evaluation of dietary ARA and DHA intakes (Forsyth, S., Gautier, S., Salem, N. Jr. Global Estimates of Dietary Intake of Docosahexaenoic Acid and Arachidonic Acid in Developing and Developed Countries. Ann. Nutr. Metab. 2016, 68, 258-67). We agree with the reviewer that the previous assumption on the sources of dietary ω-6/ω-3 disequilibrium is not presently clearly established.
- Missing to add the objective of the study at the end of the introduction
We added the objectives of the study at the end of the introduction (lines 130-133 of the revised version)
- Statistics: a. Was a sample calculated? b. Was the normality of the data analyzed?
a- Since we had no a priori estimate of the differences that we could find in microbiota analysis and measurement of expression levels of the various investigated markers, we did not calculate a sample size for our study. We estimated the number of animals needed on the previous experiences of the two laboratories involved in the study, which established an average of 10 animals are needed for the microbiota studies and 4 to 5 animals for the western blotting and quantitative RT-PCR analyses.
b- For the statistical analyses, we first checked the conformity of normality and homoscedasticity (or equality of variance of each group) before undertaking a parametric approach. If one of these assumptions is violated, a non-parametric approach was used.
- Discussion: I suggest adding weaknesses and strengths of the study, I also think a conclusion is necessary with what was mainly found
According to the reviewer’s requests, we added weaknesses and strengths of the study at the end of the discussion (lines 592-603) and a conclusion after the discussion to summarize the main findings (lines 613-622 of the revised version).
Reviewer 3 Report
Pinchaud K. et al., characterized dietary arachidonic acid (ARA) lead to low-grade colic inflammation and induced astrogliosis in the brain regardless of positive effects of dietery ARA . Although, this is an interesting manuscript, however author need to address some points to improve their study.
1. It was surprising that feeding HL diet (without ARA) for 9 weeks did not induce body weight however feeding high lipid diet even for 5 weeks induce body weight in mice is shown by many literatures. How the author can defend this?
2. Again if there is no change in body wt and food composition in all grouped mice how mesenteric adipose tissues wt.was significantly decreased in HL+ARA diet?
3. All image is not clear. I urged author to make image bold and clear in your manuscript.
4. In fig. 7, would you clarify, what is blue staining?
Author Response
- It was surprising that feeding HL diet (without ARA) for 9 weeks did not induce body weight however feeding high lipid diet even for 5 weeks induce body weight in mice is shown by many literatures. How the author can defend this?
We agree with the reviewer about the induction of obesity by high fat diets in mice. These high fat diets, which simulates human western diets, contains about 45% lipids (w/w) and mostly saturated fatty acids. But our two moderately HL-ARA and HL+ARA diets differ from the high fat diets on two points: 1- 15% fat content (w/w), 2- high amounts of mono- and polyunsaturated fatty acids (around 73%). The absence of obesity in our study is consistent with those obtained by Benoit et al. (reference 39 in the revised version), who compared the effects of moderate high fat diet (20% lipids, 70% saturated fatty acids) to those of very high fat diet (45% lipids, 70% saturated fatty acids) in C57BL/6J mice for 12 weeks. These authors observed that, while very high fat increases by 30% the total weight of the mice and by 3-fold the weight of the total fat tissues, the moderate high fat diet did not significantly modify the animal weight, nor the total fat mass, nor the weight of the various adipose tissues (mesenteric, epididymal and subcutaneous tissues) (see figure 1 of this paper). Regarding the effects of dietary unsaturated fatty acids compared to those of saturated fatty acids, Cintra et al. studied the effects of the substitution of 10% to 30% of flat seed oil (increase of polyunsaturated ω-3 fatty acids) or oleic oil (increase of monounsaturated oleic) in rats. The substitutions of mono- or poly-unsaturated fatty acids to saturated fatty acids led to reductions of rodent weight and adiposity (see figures 1 and 5. We added this paper as reference 75 in lines 556-558 of the revised version.
- Again if there is no change in body wt and food composition in all grouped mice how mesenteric adipose tissues wt.was significantly decreased in HL+ARA diet?
As we explained in discussion (lines 563-567 in the revised version), Wernstedt Asterholm et al. showed by producing mutant TNFα and suppressing the cellular response to this cytokine in adipose tissue, that the transgenic mice display reduced adiposity after 11 weeks of high fat diet feeding (35% fat w/w). They concluded that inflammation could have beneficial effects in adipose tissus by allowing expansion of this tissue in case of high fat intake. We assume that dietary arachidonic acid can alter metabolic response of mesenteric adipose tissue. It is of interest that we observe a reduction of TNFα expression levels in mice fed the ARA+HL diet compared to the animals fed the ARA-HL diet. But additional investigations are necessary to confirm this hypothesis.
- All image is not clear. I urged author to make image bold and clear in your manuscript.
We have redone all the figures in this revised version by increasing the font size and line thickness. We hope that this will satisfy the reviewer.
- In fig. 7, would you clarify, what is blue staining?
We specified in the Figure 7 legend that blue staining corresponds to the DAPI staining of cell nuclei
Round 2
Reviewer 3 Report
Authors have answered ll my questions. Now this manuscript appropraite for publication